# Modelling Endometriosis Using In Vitro and In Vivo Systems

**DOI:** 10.3390/ijms26020580

**Published:** 2025-01-11

**Authors:** Verity Black, Cemsel Bafligil, Erin Greaves, Krina T. Zondervan, Christian M. Becker, Karin Hellner

**Affiliations:** 1Nuffield Department of Obstetrics and Gynaecology, University of Oxford, John Radcliffe Hospital, Women’s Centre, Oxford OX3 9DU, UK; verity.black@gtc.ox.ac.uk (V.B.); krinaz@well.ox.ac.uk (K.T.Z.); christian.becker@wrh.ox.ac.uk (C.M.B.); 2Botnar Research Centre, NIHR Biomedical Research Unit Oxford, Nuffield Department of Orthopaedics, Rheumatology and Musculoskeletal Sciences, University of Oxford, Oxford OX3 7LD, UK; 3Division of Biomedical Sciences, Warwick Medical School, University of Warwick, Coventry CV4 7AL, UK; erin.greaves@warwick.ac.uk; 4Wellcome Trust Centre for Human Genetics, University of Oxford, Roosevelt Drive, Oxford OX3 7BN, UK

**Keywords:** endometriosis, model, in vitro, in vivo, animals, endometrium, tissue culture

## Abstract

Endometriosis is a chronic inflammatory condition characterised by the presence of endometrium-like tissue outside the uterus. Despite its high prevalence and recent advances in molecular science, many aspects of endometriosis and its pathophysiology are still poorly understood. Previously, in vitro and in vivo modelling have been instrumental in establishing our current understanding of endometriosis. As the field of molecular science and the advance towards personalised medicine is ever increasing, more sophisticated models are continually being developed. These hold great potential to provide more intricate knowledge of the underlying pathophysiology and facilitate investigations into potential future approaches to diagnosis and treatment. This review provides an overview of different in vitro and in vivo models of endometriosis that are pertinent to establishing our current understanding. Moreover, we discuss new cross-cutting approaches to endometriosis modelling, such as the use of microfluidic cultures and 3D printing, which have the potential to shape the future of endometriosis research.

## 1. Introduction

Endometriosis is a common oestrogen-dependent, chronic inflammatory disease associated with debilitating pelvic pain and reduced fertility. It is defined as the presence of endometrium-like tissue, such as epithelial and stromal cells, outside the uterine cavity [1]. It is estimated that 10% of women of reproductive age are affected, totalling at least 190 million worldwide [2]. Traditionally, endometriosis diagnoses occur via laparoscopic surgery, and it is therefore not unsurprising that on average, women have to wait 8–12 years to receive a diagnosis. Despite significant advances in our understanding of the pathogenesis of endometriosis since its initial description in the early 20th century, the exact mechanisms behind this chronic pain condition remain elusive [3]. One of the most referred to theories of endometriosis pathogenesis is Sampson’s theory of retrograde menstruation. It proposes that menstrual tissue is refluxed into the peritoneal cavity during menstruation. A combination of immunological dysfunction and maladaptive transcriptional changes then leads to lesion implant, neovascularisation, and proliferation into these new microenvironments [Figure 1]. Many types of studies, from single-cell experiments to primate models, have been conceived to try and further understand this disease as well as identify potential therapeutic targets. This review focuses on some of the important in vitro and in vivo models of endometriosis and discusses their advantages and limitations whilst also looking ahead to promising new research tools under development.

## 2. Primary Endometrial Samples and Cell Lines

All biological models of endometriosis require a supply of endometrial or endometriotic cells, either from primary samples or established cell lines. Both have advantages and limitations regarding their use, storage, and properties. Cells sourced directly from the clinically relevant tissue and subsequent establishment of cell lines have been at the forefront of endometriosis research. While technical challenges of their storage and invasive nature of the collection methods, primary cells provide the closest impression of endometriosis in milieu. Cell lines, on the other hand, are easier to maintain and can be used infinitely to examine different biological questions.

### 2.1. Primary Cells

Primary cells derived directly from eutopic and ectopic endometrium provide a useful, physiologically relevant tool to mimic the in vivo processes occurring during endometriosis pathogenesis. They largely maintain their normal cell morphology and specific surface markers, enabling purity screening, and thus minimise the variability of data between groups, enabling large-scale collaborations [4]. The cells are usually extracted from surgical biopsies and are subsequently isolated and cultured. Recently, significant efforts have occurred to try and standardise these procedures with the aim of improving study reproducibility and cross-model validation [5,6].

Endometriotic epithelial and stromal cells specifically have widely been utilised to study various aspects of endometriosis, including inflammation, angiogenesis, and proliferation. Stromal cells are more commonly used due to their ease of culturing and ability to be routinely passaged up to 20 times owing to their high recovery and viability rates even following cryopreservation [7,8]. Epithelial cells, however, are highly differentiated, are strongly adhesive, and must maintain polarity in order to sustain their apical/basolateral morphology. Therefore, longer term storage and the use of primary epithelial cells in in vitro studies has previously been limited. However, recent genetic studies have theorised that these epithelial progenitor cells likely play a key role in establishing ectopic nascent glands. The recruitment of polyclonal stromal cells is hypothesised to be a key step in the formation of infiltrating endometriotic lesions [9]. New findings like this highlight the need for the development of more robust extraction and storage methods of all types of endometriotic cells to enable high-quality studies of these interactions on a cellular basis.

#### 2.1.1. Accessing Primary Endometrial Cells

Endometrium consists of glandular and luminal epithelial and stromal cells, which are closely spatially and functionally related despite possessing distinct molecular and genetic features.

Eutopic endometrium can usually be obtained through minimally invasive means, e.g., either during surgery or in an outpatient/office setting using brushes, suction, or scraping devices. Alternatively, intra-operative endometrial curettage or direct post-operative excision/scraping of endometrium from hysterectomy specimens can also be used. The tissue yield and quality vary according to the sampling device, technique, operator experience, and, most importantly, the menstrual cycle stage and hormonal therapy status [10]. The use of menstrual blood-derived cells has emerged in recent years as an alternative, non-invasive method for mimicking endometriosis in vitro [11,12].

Ectopic endometrium is often obtained during laparoscopy when lesions are excised or ovarian endometriotic cysts are removed [13,14]. The amount of ectopic endometrium collected varies, as peritoneal lesions are often small and also shrink once excised due to intrinsic tension. Ovarian endometriotic cysts commonly consist of a fibrotic capsule of varying thickness, which is lined by a thin layer of endometriotic tissue. As a result, the quality and quantity of endometriotic cells derived from such cysts may be reduced.

#### 2.1.2. Primary Cell Separation

For subsequent study, it is often necessary to separate endometriotic stromal and epithelial cells. This is achieved most commonly by enzymatic tissue digestion, usually with collagenase, to prepare a cell suspension that can be plated onto tissue culture dishes [15]. Stromal cells will attach easily to conventional tissue culture plates, whilst epithelial cells are likely to float initially and thus can be separated [13]. It is also possible to filter the digested tissue through a cell strainer of differing mesh sizes with the aim of collecting a single-cell suspension comprising endometrial epithelial and stromal cells and leukocytes [16]. Particular cell types of interest can now be obtained by sorting via FACS (Fluorescent Activated Cell Sorting) and/or MACS (Magnetic Activated Cell Sorting) using relevant cell surface markers [17]. For example, endometrial epithelial progenitor cells can be isolated using N-cadherin and EpCAM [18], while endometrial stromal cells can be isolated with CD90, CD73, CD105, and HLA I positive expression and absent CD45 expression [19]. An overview of surface markers for the selection of endometrial stem cells and stromal cells can be found in the review article by Gargett et al. [20]. Additionally, leukocyte subpopulations, such as lymphocytes and macrophages, can further be isolated by density gradient centrifugation [21]. These advances in cell sorting have enabled more reliable and quicker processing of biological samples and thus have facilitated more rapid advances in endometriosis modelling.

#### 2.1.3. Limitations of Primary Cells

Although it is generally thought that primary cells are phenotypically and genotypically similar to their tissue of origin, the process of isolation and culturing disturbs normal cellular crosstalk and thus results in cellular adaption. Transcriptome studies have shown that during this process, significant shifts in subpopulation dynamics can be observed in response to the new in vitro conditions. The selection pressure subsequently leads to the development of subpopulations with altered properties, such as increased growth rates or trypsin sensitivity [22,23]. Therefore, primary cells are still not exact replicates of endometrial cells in vivo, and the more passages that they are subjected to, the less closely they will mirror the original cells. Secondly, certain cell types require specific culture conditions. Numerous studies [24,25,26,27] have developed customised culture media to enhance primary cell survival, particularly that of epithelial cells, which have been previously found to enter a senescent state within two weeks of initial culture [28]. However, a consensus as to the optimum culture media is not clear cut. Finally, it has been widely documented that primary endometrial cultures often have significant levels of heterogeneity caused by erroneous collection of other cell types, with some studies estimating contamination rates between 1 and 5% [29,30,31]. This heterogenicity is particularly important for slowly proliferating cells, such as epithelial cells, as contamination with rapidly proliferating cells can significantly impact the sample composition. Thus, a careful analysis of the sample composition is required to ensure the authenticity of any results found.

### 2.2. Cell Lines

Cell lines are frequently used in endometriosis research because of their high proliferation rates, easy culture, and simple transfection. Although there is clear evidence that immortalisation of primary cells and serial passaging leads frequently to alteration in genotype and phenotype [32], cell lines are still key cost-effective tools for basic endometriosis research. Within the last two decades, several cell lines have been established from patient-derived endometriosis lesions and endometrium (Table 1), but the overall number of cell lines has remained small. Peritoneal and ovarian lesions were used to establish cell lines that are used to investigate genomic, invasion, and proliferation properties of endometriotic cells, among other uses. hEM15A cell line is the first and only eutopic endometrium-derived cell line from an endometriosis-presenting individual.

#### 2.2.1. Advantages of Using Cell Lines

Because collecting primary cells for detailed studies would often not be feasible, cell lines that can be maintained for longer periods of time and undergo multiple passages frequently represent a compromise. Established cell lines exhibit much less inherent variability in comparison to primary cells. Several studies have investigated whether immortalised cell lines retain similar transcriptomes to primary samples, particularly in certain genes involved in steroid biosynthesis, angiogenesis, and extracellular matrix degradation—all characteristics that are important in the establishment and persistence of endometriosis in vitro [39]. Semi-quantitative RT-PCR studies conducted by Banu et al. showed that immortalised ectopic epithelial and stromal cell lines retained characteristic alterations in their disease-linked gene expression. The turnover of the extracellular matrix is regulated by a number of matrix metalloproteinase (MMP) and tissue inhibitor of metalloproteinase (TIMP) genes, and it has been shown previously that some of these genes are aberrantly expressed in endometriotic tissues [40]. Banu et al. also demonstrated that MMP2 and MMP9 were upregulated in a range of immortalised endometriotic epithelial and stromal cell lines, which mimics primary cultures and supports the use of these cell lines as accurate model systems.

#### 2.2.2. Limitations of Cell Lines

Despite the number of advantages of using cell lines for research, there are also several limitations that should be acknowledged. Immortalised cell lines are composed of only one cell type (epithelial or stromal cells) and therefore represent a small, specific subset of the disease phenotype. They often exhibit features of undifferentiated cells, which do not accurately represent normal cell behaviour seen in vivo. Furthermore, when cells are cultured, they lack their natural environment and interactions with other cell types—factors that may be critical to the hypothesis being tested [41]. Also, long-term cultivation and repeated serial passages of the cell lines increase the risk of losing tissue-specific properties and introduce more genetic instability over time [42]. Korch et al. analysed DNA microsatellite short tandem repeats of 10 endometrial cell lines commonly used in preclinical endometriosis studies. This showed significant levels of duplication and loss of integrity of these endometrial cell lines. Therefore, it is crucial that cell lines should be monitored and authenticated to ensure any significant genotype–phenotype changes are recognised and that results drawn are still meaningful in the context of the disease biology and development.

## 3. In Vitro Modelling

In vitro models enable the study of the pathophysiology of endometriosis on a cellular and molecular level. In the past, attachment and two-dimensional invasion models that utilise either primary samples or immortalised cell lines have provided insights into the mechanisms involved in the cell–cell interactions that drive endometriosis [43]. With the advent and development of cutting-edge genomic sequencing techniques, our understanding of the human endometrium and its dynamic regulation throughout the menstrual cycle has significantly improved [44,45]. Furthermore, as more sophisticated genomic techniques have been developed, such as single-cell RNA sequencing, the ability to investigate endometriosis-associated genomic changes has improved significantly, furthering our knowledge of the underlying genetic mechanisms of the disease and improving the quality of endometriosis models. More recently, the use of three-dimensional (3D) modelling has allowed for more complex systems to be formed, which demonstrate more realistic cell–cell interactions in terms of polarity, contact inhibition, and angiogenesis [46]. These 3D models therefore mimic natural tissues and organs more reliably than cells grown in conventional monolayer cultures [47,48].

### 3.1. Single-Cell Studies

Single-cell studies have facilitated significant improvements in our understanding of the molecular pathogenesis of endometriosis and hence have enabled more translatable in vitro models to be developed. For example, these studies can generate spatiotemporal cell atlases and allow for investigations of the transcriptomes of both eutopic and ectopic endometrium. A study by McKinnon et al. investigated transcriptomic changes observed in over 33,000 cells from ten women with endometriosis and nine controls [49]. They used a combination of Louvain clustering and immunofluorescence techniques to determine that differential gene expression existed between the two groups in genes regulating cell growth and the cell cycle. They also noted that purified samples of ectopic lesions contained mesenchymal and stromal fibroblast cells, indicating that divergent differentiation is likely occurring during endometriosis. Similar transcriptomic profiles were found in the menstrual effluent of individuals with endometriosis, providing evidence that the use of this material may present an accurate sample of tissue for further in vitro modelling, which does not require invasive surgical acquisition [50]. Other single-cell analyses have demonstrated fundamental differences between eutopic endometrium and ectopic lesion microenvironments, which may contribute to disease progression and thus need to be considered when developing endometriosis models [51,52]. Tan et al. utilised scRNA-Seq in conjunction with hyperplexed antibody imaging techniques to form comprehensive atlases of peritoneal and ovarian lesions extracted from endometriosis patients. They found significant changes in the vascular and immune transcriptional profiles within both types of lesions. This built upon previous evidence suggesting that myeloid-derived immune cells play a key role in this disease by demonstrating that macrophages present in the lesion microenvironments exhibit an immunotolerant phenotype, whilst dendritic cells exhibited higher levels of the immunoregulatory receptors VSIG4 and MRC1 [53,54,55]. Despite the insight provided by these studies, existing single-cell analyses remain underpowered and therefore are unable to accurately adjust for potential confounding factors and inter- and intra-individual variations.

### 3.2. Two-Dimensional (2D) Cell Culture Models

Two-dimensional cell culture models primarily consist of monolayers of cells cultured either in a flask or petri dish. Their wide-spread use can be attributed to the low cost of their maintenance, the rapid time of culture formation, and high reproducibility. Two-dimensional models have been pivotal in the study of the basic pathogenesis of endometriosis. For example, in conventional monolayer cell culture models, the functional relationships between certain hormones and endometriotic cells have been studied. Lee et al. [56] studied the interaction of prostaglandin signalling and endometriotic cells, as well as its role in cellular proliferation. They showed that the selective inhibition of specific prostaglandins induced cell cycle arrest at different points in both endometriotic epithelial and stromal cell lines and thus suggested that this could lead to the development of non-oestrogen-dependent treatment options in the future. Furthermore, the inflammatory and immune components of endometriosis have been investigated using 2D cell models, as they enable cell lines to respond to certain immune regulators in a measurable fashion [57]. In 12-Z cells, interleukin-33 was shown to induce the production of pro-inflammatory cytokines, such as chemokine ligand 1, aligning with previous evidence, which demonstrated that certain chemokine receptors are overexpressed in endometriotic lesions [58].

The development of co-culture models has significantly improved our understanding of cell–cell communication, particularly that of endometriotic epithelial and stromal cells. Chen et al. established a two-chamber co-culture system in which the two cellular compartments are cultured in different media or conditions with no direct cell–cell contact [59]. They observed the formation of a polarised, tight epithelial barrier resembling the in vivo state of endometriotic tissue. However, these 2D models lack the intricate and dynamic interactions between cells and their matrix as well as possess non-physiologically uniform oxygen and nutrient gradients. This likely contributes to their poor reproducibility and thus poor predictability of in vivo drug efficacy [60].

### 3.3. Three-Dimensional Cell Culture Models

Three-dimensional cultures are artificially created environments that more closely mimic in vivo conditions by enabling the growth and interaction of cells in a multidirectional fashion. They enable the formation of 3D cell aggregates either surrounding a scaffold or matrix or in a scaffold-free manner, such as in suspension. They have greater stability, have longer potential life spans, enable aggregate formation in non-turbulent environments, and demonstrate genotypes more closely mimicking that of the in vivo tissue [61].

#### 3.3.1. Three-Dimensional Matrices

Cell studies utilising 3D matrices have been found to differ in their gene expression profiles compared to those cultured in 2D environments, often better mirroring the genotypes documented in vivo [62]. This likely plays a role in the different behaviour of cells in each culture, including their growth, invasion, and migration properties. To model the biology of ovarian endometriosis, Brueggmann et al. established endometrial epithelial cell lines EEC16 and 12Z as a 3D spheroidal model [46]. The models showed differential expression of endometriosis-associated genes compared to ovarian epithelial cells taken from individuals without endometriosis and those in 2D monolayer cultures. They also found that when cultured in this manner, they more closely resemble the geometry and elasticity seen in vivo. Further endometriotic spheroids have been developed utilising both immortalised endometriotic epithelial cells and uterine stromal cell lines to enable a direct comparison of transcriptomic profiles [63]. These studies demonstrated that inflammation pathways were upregulated in endometriotic epithelial spheroids compared to uterine stromal spheroids, mimicking previous profiles extracted from baboon endometriotic lesions. The resemblance that models have to in vivo environments is crucial when screening potential therapeutics where geometry and structural likeness are key, such as small molecule inhibitors, as well as identifying and avoiding potential unwanted side effects, which would not be observed in a 2D monolayer culture. The importance of the similarity in genotype between 3D cultures and in vivo cells was exemplified by studies investigating statins as a therapeutic approach to endometriosis due to their anti-angiogenic activity [64,65,66]. Furthermore, Lv et al. constructed a 3D Matrigel-based culture system to explore how PTEN impacted vessel growth mimicry in endometriotic lesions. The group demonstrated that in areas of high PTEN expression within the 3D cultures, angiogenesis was supressed, suggesting that PTEN gene therapy could be utilised in the future as a tool to limit endometriotic lesion growth [67].

Collectively, these 3D models demonstrate some of the advancements that have been made in understanding the underlying biological pathways which dictate endometriosis development.

#### 3.3.2. Self-Assembling Organoids

Within recent years, significant effort has been put into developing more sophisticated in vitro and ex vivo models that simulate in vivo function, physiology, and the crosstalk occurring within the female reproductive tract beyond simple 3D matrices [68,69]. Organoids are a growing type of 3D culture that contain progenitor and differentiated cells to resemble the origin tissue in a self-organising fashion. Normal endometrial cells have been used to establish 3D endometrial organoids that were hormone responsive, were genetically stable, and can be passaged long term [70,71]. A scaffold-free, 3D organoid containing endometriotic epithelial and stromal cells was recently produced by Wiwatpanit et al. During self-organisation, a stromal cell core and epithelial cell outer layer was formed, which closely mimics morphologies of endometriosis lesions [71]. Although organoids have limited nutrient exchange, by combining these in vitro models with spinning bioreactors or by culturing them with endothelial cells, this limitation can be significantly lessened [72]. Furthermore, previously, these studies were limited by the invasive process required to gather tissues, and thus, studies were confined to more advanced disease states. However, recently, studies have shown the potential for organoids developed from menstrual fluid that mimic the transcriptome, receptor profiles, and other physiological parameters seen in current organoids produced from invasive surgical procedures [73]. For example, Filby et al. demonstrated that organoids produced from menstrual fluid from women at different stages of disease showed graded clustering of genes regulating epithelial secretory and androgen receptor signalling, a finding which may significantly aid the advancement of minimally invasive precision medicine. These self-assembling organoids, which mimic functional and molecular signatures of endometrium, will allow for further interrogation of endometriosis pathogenesis and novel drug therapies.

## 4. In Vivo Modelling

Occurring within living organisms, most often animals, in vivo models more closely mimic the biological processes occurring in humans. Primates are known to spontaneously develop endometriosis, and therefore, primate models most closely resemble the disease and are ideal to study. However, rodent models are more cost-efficient and abundant than primate models. Whilst not all aspects of the human disease can be replicated by currently available animal models, they have been indispensable tools for the improvement of early diagnosis and treatment.

### 4.1. Non-Primate Models

The low cost of non-primate models, easy handling, and the possibility of genetic manipulation using knockout and transgenic animals have made non-primate models very popular [74]. Although rat and mouse models in particular have improved significantly in recent years, these models still have some major disadvantages [75]. The most crucial drawback is that rodents do not menstruate spontaneously and therefore do not spontaneously develop endometriosis. Most rodents remodel their endometrium during an oestrous cycle, a physiological difference likely due to the wide phylogenetic gap between non-primates and humans. To overcome this, the initiation of endometriosis-like lesions in these models has usually required the introduction (via stitching or injection) of uterine tissue fragments to the recipient peritoneum or through significant genetic manipulation [76]. Depending on which tissues are used for this, rodent models can be separated into two categories: homologous and heterologous models. Homologous, or allograft, models involve the use of uterine tissue from the same animal (autologous) or from a synergistic animal. Heterologous, or xenograft, models use human tissue and immunosuppressed rodents such as athymic or NOD/SCID mice.

#### 4.1.1. Homologous Models

Homologous models of endometriosis are most commonly formed through the surgical transplantation of uterine tissue fragments into an ectopic location. This type of model presents many advantages, including their non-dependence on human tissue, model immunocompetence, and the low risk of implant rejection. The ectopic uterine fragments also mimic histological features of human counterparts [77].

This basic premise has been expanded upon within the past years to improve the likeness of these models to human endometriosis. Karen Berkley’s group published an autologous rat model of endometriosis that involved uterine fragments being sutured to the mesenteric cascade of the intestine, the peritoneal wall, and the ovary [78]. They also were first to demonstrate vascularisation and innervation of these lesions, leading to vaginal hyperplasia and oestradiol-dependent vaginal nociception [79,80]. Further modifications to this model include the grafting of autologous uterine tissue onto the gastrocnemius muscle for electrophysiological recording of sensory neurons, which innervate ectopic tissue and have been hypothesised to play a role in endometriosis-associated pain [81,82]. However, although useful for the exploration of peripheral nerve function and pro-nociceptive molecules, this model does not recapitulate the interactions that occur between the peritoneal lining and endometrial fragments.

Over the past two decades, numerous genetically modified mouse models have been developed to enable modelling of the initiation and pathogenesis of endometriosis without the need for surgical intervention [83]. Dinulescu et al. utilised Cre-lox technologies to develop the first genetically modified mouse model, where they showed that de novo, benign ovarian endometriosis-like lesions could be stimulated via oncogenic K-ras induction of the ovarian epithelium [74]. Wilson et al. took this further, using the lactotransferrin-Cre allele to show that ARID1A and PIK3CA mutant endometrial epithelium exhibited locally invasive properties, which may enable initial retrograde menstruation hypothesised to drive the propagation of endometriosis [84].

Deep endometriosis is a subtype of the disease often underrepresented in mouse models despite its clinical severity. The first successful mouse model of deep endometriosis was established in 2019 and provided support for the theory that deep lesions act like wounds, undergoing repeated tissue injury and attempted repair [85]. This highlighted the potential for microenvironment-based interventions, which could alter these biochemical processes and ease the clinical burden of deep endometriosis.

Despite the many advantages of homologous rodent models of endometriosis, there are several limitations, which must be considered. In surgically induced mouse models of endometriosis, both endometrium and myometrium are often implanted together [84]. However, a number of studies have demonstrated ways to minimise this, such as through the use of bioluminescence imaging [86]. Furthermore, a study by Dorning et al. utilised this technique to compare four different “menses” models of induced endometriosis. They found that the four similar models exhibited significant variability in the underlying “disease” mechanisms, which may indicate why therapeutic experiments often have low reproducibility, both between homologous models and during clinical application [87]. However, this diversity in pathological mechanisms may be utilised to improve the reproducibility of results from pre-clinical models to clinical trials, as by using multiple models in a pre-clinical environment, one can more accurately simulate the heterogeneity of disease phenotype seen clinically. The realistic modelling of menstrual shedding and transport has also been limited in early surgical mouse models. The discovery that the spiny mouse (Acomys cahirinus) undergoes spontaneous decidualisation presented significant potential for the development of more translatable models. The spiny mouse also shares numerous physiological similarities with primates, including uterine secretion of repair and inflammatory proteins and similar stages of endometrium breakdown, repair, and rebuilding [88]. To better reproduce the phenomenon of retrograde menstruation, a number of models have been utilising mouse “menstrual” donor endometrial tissue. Endometrial shedding and repair mimicking the human menstrual cycle can now be induced by hormonal manipulation and artificial decidualisation [89]. For example, a syngeneic immunocompetent model whereby donor ‘menstrual’ material was injected into the peritoneal cavity of recipient mice developed lesions that phenocopy those recovered from women in histological appearance; steroid receptor expression; and blood vessel, nerve fibre, macrophage infiltration, and altered sensory behaviour [87,90,91,92,93]. Since endometriosis-associated pain is an important measurement when testing potential therapeutic approaches, models like these that enable pain evaluation are crucial to obtaining translatable results in humans. However, a recent systematic review highlighted the need for models that utilise measures of pain beyond simple reflex tests [94].

#### 4.1.2. Heterologous Models

Heterologous models utilise immunocompromised rodents, enabling the transplantation of human tissue xenografts, thus allowing investigations to occur in “humanized” models. Both human endometrial tissue and endometriotic cell lines have been successfully implanted either subcutaneously or intraperitoneally into a range of immunodeficient mice to prevent rejection [95,96,97]. Many initial studies utilising athymic mice, which lack mature T cells, demonstrated the ability to implant human endometrial tissue as well as sustain the development of glandular endometrial lesions and vascular networks [98]. However, these models quickly become non-functional due to the infiltration of lymphocytes. Therefore, the use of other immunocompromised mice, such as SCID or NOD/SCID mice, have since been utilised [99,100]. However, such models compromise research exploring the role of the immune system in endometriosis—a factor that has increasingly been recognised as a key driver of the pathology. Therefore, the relevance of these models to understanding the pathology of inflammatory-based diseases such as endometriosis may be limited, although their important role in investigating potential therapeutic still remains [101,102].

### 4.2. Non-Human Primate Models

Primate models have been an excellent tool in the research of the pathogenesis and treatment of endometriosis due to both the close phylogenetic relatedness and the reproductive anatomical and physiological similarities [103]. Nevertheless, the use of non-human primate models has been limited, primarily due to the ethical considerations and associated maintenance costs.

#### 4.2.1. Spontaneous and Induced Models

The spontaneous development of endometriosis with lesions histological and temporally comparable to humans has been reported in 11 species of non-human primates [104,105,106,107]. The artificial induction of endometriosis has been conducted through various methods, such as the initial diversion of menstrual flow into the abdomen through the repositioning of the cervix in rhesus monkeys [108]. Subsequently, many other approaches have been employed such as occlusion of the uterine cervix to alter menstrual flow or the transplantation of homologous endometrial tissue ectopically into the peritoneal cavity [109,110,111]. Intraperitoneal inoculation of endometrium successfully led to the development of lesions within baboons, which were similar to those developing in spontaneous models, thus providing a robust way of inducing the disease in far shorter time periods.

#### 4.2.2. Key Areas of Research Utilising Non-Human Primate Models

Because of the anatomical and physiological similarities to humans, non-human primate models have been particularly useful when modelling endometriosis. Both rhesus monkeys (Macaca mulatta) and cynomolgus monkeys (Macaca fascicularis) have been pivotal in the investigation of the genetic epidemiology of endometriosis, particularly as colonies of captive rhesus monkeys have smaller gene pools and greater genetic homogeneity and are therefore likely to possess unique or high-frequency risk alleles. A study of autopsy records from a colony of rhesus monkeys showed that elevated oestradiol levels was a significant risk factor for endometriosis [112]. Furthermore, familial clustering of endometriosis has been documented in humans [113] and rhesus monkeys [114]. Recently, DNA sequencing studies have found that deleterious coding variants within the gene NPSR1 were overrepresented in individuals with endometriosis; a finding that was replicated in a population of rhesus macaques with spontaneous endometriosis [115]. This highlights the relevance of non-human primate models in current and future genomic explorations of endometriosis.

As these animals live in controlled environments, it enables a more accurate retrospective analysis of environmental contributors to endometriosis development. For example, both dioxin and polychlorinated biphenyls were shown to cause a dose-dependent increase in both the incidence and severity of endometriosis occurring in rhesus monkeys and rats [116,117,118].

The role of the immune system in endometriosis development and progression has been investigated in non-human primate models. In baboons, it was shown that inflammation caused by spontaneous retrograde menstruation and an experimental intra-pelvic injection of endometrium were associated with increased peritoneal fluid volume and an increased concentration of white blood cells and inflammatory cytokines. The coexistence of endometriosis and peritoneal inflammation was thus concluded to be a potential target for novel anti-inflammatory therapeutic options. D’Hooghe et al. demonstrated that the use of r-HTBP-1 to neutralise the pro-inflammatory cytokine TNF-a inhibited the development of endometriosis lesions and adhesions in baboon models [119]. The administration of Etanercept, which neutralised TNF-a activity, to female baboons with spontaneous peritoneal endometriosis led to decreased lesion number and surface area. Sadly, this did not translate directly into humans, as a small, randomised placebo-controlled trial found that treatment with an anti-TNF-a monoclonal antibody had no effect on either pain or volume and appearance of the endometriotic nodules [120].

## 5. Cross-Cutting Models

### 5.1. Three-Dimensional Microfluidic Cultures

There have recently been a number of significant advances in endometriosis research. One such breakthrough has been the application of microfluidics to biological modelling of whole tissue and organs systems alongside the development of “Organs-on-chip” (OoC) 3D cell cultures. Microfluidic platforms such as Solo-MFP, Duet-MFP, and Quintet-MFP utilise either pneumatic or embedded electromagnetic actuation technology to control fluid movement throughout a system that mimics circulation occurring in vivo. This allows for the use of blood-like fluid, which can transport hormones and drugs between different tissues within a 3D culture. These techniques have enabled the development of a novel in vitro model of reproductive tract tissues alongside peripheral organs, such as ovaries, ovarian follicles, fallopian tubes, and endometrium, facilitating organ-to-organ hormonal signalling integration [121]. When studying a disease reliant on timely endocrine loops, such as those controlling the menstrual cycle, this type of model accuracy is crucial. The decellularized uterine scaffolds used within the microfluidic system enabled the previously difficult growth and maintenance of both epithelial and stromal endometrial cells for 28 days. Going forward, it is expected that the system can be refined further to be inclusive of subsidiary cell types such as immune cells [122,123].

A vascularised endometrium-on-a-chip 3D model engineered by Ahn et al. demonstrates the feasibility of these novel inventions to not only study the disease but also perform drug screening and discovery [124]. This model comprises three distinct layers of the endometrium: epithelial cells, stromal fibroblasts, and endothelial cells, all cultured within a 3D extracellular matrix. This study successfully replicated endometrial vasculo-angiogenesis and the complex hormonal responses characteristic of the menstrual cycle. Most recently, Busch and colleagues developed hormone-responsive miniaturised models of the uterine wall by incorporating patient-derived endometrial and myometrial cells [125]. Three-dimensional microfluidic cultures will no doubt expedite the understanding of the pathophysiology of endometriosis and the interactions between endometrial and other cell types.

### 5.2. Three-Dimensional Printing

The use of 3D printing is emerging as a useful tool for the development of microfabricated constructs, which closely mimic the organisation and architecture of naturally occurring tissues [126]. Laronda et al. used 3D printing to create a gelatinated scaffold for the culture of ovarian murine follicles, which could then be implanted into ovariectomised mice, demonstrating that this restored ovarian function in vivo and live births [127]. The ability to create models utilising functional in vivo implants may enable more accurate and controllable models of human endometriotic lesions in animal models going forward. However, it has been shown that typical 3D printing protocols and resins may not be appropriate for reproductive studies due to their association with oocyte degeneration [128].

### 5.3. Menstrual Blood Derived Stromal Cell Models

A novel in vivo model of endometriosis was created by Zhang et al. that utilises menstrual blood-derived stromal cells (MenSCs) [11]. This model was created based upon a range of contemporary studies highlighting the difference between eutopic endometrium and other endometrial tissue [129,130,131]. Endometrial progenitor cells, which are easily sampled via menstrual blood, have significant transformation and proliferation capacity, thus making them ideal for heterologous in vivo modelling. Nikoo et al. demonstrated that there are distinct molecular differences between MenSCs taken from healthy individuals and those with endometriosis [132]. MenSCs from individuals with and without endometriosis implanted into nude mice found an 80% lesion formation rate when using cells from individuals with endometriosis compared to healthy controls. This new approach to in vivo modelling of endometriosis has significant potential to improve the translatability between animal models and humans and improves the ease of sample collection.

## 6. Summary

In vitro and in vivo models of endometriosis are of great value for the evaluation of the pathophysiological mechanisms underlying the development of this omnipresent gynaecological disease. There are several cell lines derived from endometrium and endometriosis lesions, which continue to prove useful for basic research. For further insight into the characteristics of endometrial tissue in physiological and pathophysiological conditions, primary cells are of high importance. Recent advances in culturing methods, such as 3D microfluidic cultures and organoids, are likely to assist and expand endometriosis research in the future. Organ-on-a-chip models, for example, have been shown to replicate the functional responsiveness of endometrium, with vascularisation and immune microenvironment achieved to some extent. Furthermore, cells derived from the menstrual blood, which largely retain their molecular signature, provide an easily and repeatedly accessible source to study endometriosis and other endometrial pathologies.

On the other hand, for in vivo modelling of endometriosis, rodents and primates have been extensively used. Quite often, rodents are the preferred choice due to low maintenance costs, easy access, and less ethical issues surrounding their use. However, none of these models absolutely replicates all aspects of the disease in humans. Advantages and limitations should be thoroughly considered when deciding which particular model system is to be used for a targeted evaluation of scientific questions. Recent advances in in vivo modelling strategies, such as the use of 3D printing, microfluidics, and menstrual blood-derived stromal cells, have begun to yield positive results and thus may lead to the development of more sophisticated models of endometriosis going forwards. These advances could aid in improving both diagnostic processes and therapeutic approaches to endometriosis clinically.

## Figures and Tables

**Figure 1 ijms-26-00580-f001:**
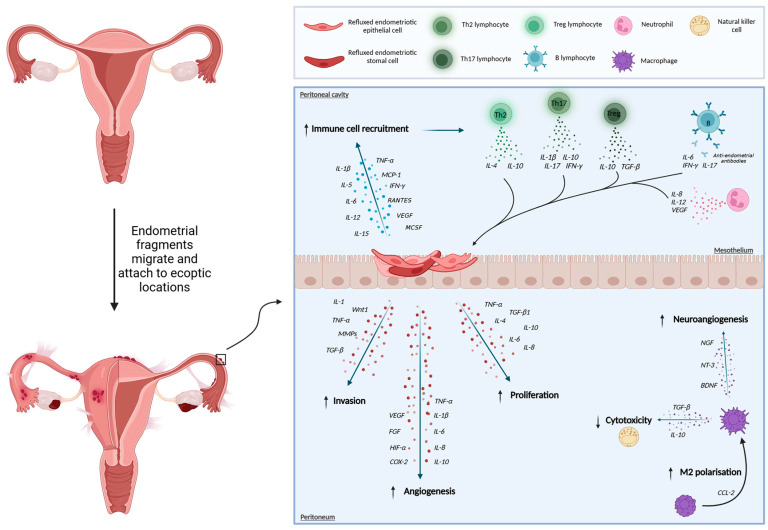
A schematic representation of an endometriotic lesion and some of the key cellular factors which are involved in the immunopathogenesis of the disease. Abbreviations: IL-, interleukin-; TNF-a, tumour necrosis factor alpha; MCP-1, monocyte chemoattractant protein-1; IFN-y, interferon gamma; VEGF, vascular endothelial growth factor; MCSF, macrophage colony-stimulating factor; TGF-b, transforming growth factor beta; MMPs, matrix metalloproteinases; FGF, fibroblast growth factor; HIF-a, hypoxia-inducible factor 1; COX-2, cyclooxygenase 2; NGF, nerve growth factor; NT-3, neurotrophin 3; BDNF, brain-derived neurotrophic factor.

**Table 1 ijms-26-00580-t001:** List of commonly used cell lines derived from endometriotic lesions to model endometriosis in vitro.

Cell Type	Cell Line ID	Tissue of Origin	Primary Research Area	Reference
Endometriotic epithelial cells	FbEM-1	Peritoneal lesion	Genomic studies	[33,34]
	EEC 145T	Peritoneal lesion	Invasion studies	[35]
	10B, 10Z, 11Z, 11E, 12Z, 33Z, 39Z, 42B, 45Z, 49Z, 50Z, 108Z	Peritoneal lesion	Invasion, proliferation, apoptosis angiogenesis and inflammation studies	[36]
	EMosis-CC/TERT1, EMosis-CC/TERT2, EMosis-E6/E7/TERT1, EMosis-E6/E7/TERT2	Ovarian lesion	Transcriptomic studies, NOTCH signalling, neoplastic transformation	[37]
	EEC16-TERT	Ovarian lesion	Transcriptome analysis	[37]
	hEM5B2	Ovarian lesion	Cell line establishment	[38]
Endometriotic stromal cells	3, 4, 9-4Z, 9-8Z, 17B, 18B, 20B, 22B, 25Z, 40Z, 55Z, 57Z-T1, 57Z-T2	Peritoneal lesion	Proliferation, apoptosis, inflammation, angiogenesis and invasion studies	[36]
	hEM15A	Endometrium	Cell line establishment	[38]

## Data Availability

Data sharing is not applicable.

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
