# Peer review of "Modelling Endometriosis Using In Vitro and In Vivo Systems"

_ijms, 2025, doi:10.3390/ijms26020580_

Round 1
Reviewer 1 Report
Comments and Suggestions for Authors
-The abstract effectively summarizes the key points but could be made more concise. Consider removing phrases that may be redundant, such as "which impacts millions of women worldwide," as the prevalence is stated later.
-The introduction provides a solid overview of endometriosis. However, including more recent statistics or studies from the last few years could strengthen the context and relevance of the topic.
Grammatical Mistakes
Despite its high prevalence and recent advances in molecular science, many aspects of endometriosis and its pathophysiology remain poorly understood. "Consider rephrasing For example: "Despite its high prevalence and recent advances in molecular science, many aspects of endometriosis and its pathophysiology are still poorly understood."
"As the field of molecular science and the advance towards personalized medicine is ever increasing, more sophisticated models are continually being developed which provide more intricate knowledge of the underlying pathophysiology and facilitate investigations into potential future approaches to diagnosis and treatment." This sentence is lengthy. Consider splitting it into two sentences or adding a comma before "which":
"This review provides an overview of different in-vitro and in-vivo models of endometriosis which have been pertinent to establishing our current understanding." Change "which have been pertinent" to "that are pertinent" for better tense consistency.
"Moreover, we discuss new cross-cutting approaches to endometriosis modelling, such as the use of microfluidic cultures and 3D printing, which will hopefully shape the future of endometriosis research." The phrase "which will hopefully shape" could be more assertive. Consider changing it to "which have the potential to shape."
"Endometriosis is a common oestrogen-dependent, chronic inflammatory disease associated with debilitating pelvic pain and reduced fertility in women during their reproductive years." The phrase "in women during their reproductive years" is somewhat redundant given the context. It could be simplified to "associated with debilitating pelvic pain and reduced fertility."
"Despite its high prevalence and recent advances in molecular science, many aspects of endometriosis and its pathophysiology remain poorly understood." Consider rephrasing for clarity, e.g., "Despite its high prevalence and recent advances in molecular science, many aspects of endometriosis and its pathophysiology are still poorly understood."
"Previously, in-vitro and in-vivo modelling has been instrumental in establishing our current understanding of endometriosis." Change "has been" to "have been" to match the plural subject.
"This review provides an overview of different in-vitro and in-vivo models of endometriosis which have been pertinent to establishing our current understanding." Use "that" instead of "which" for defining clauses, e.g., "models of endometriosis that have been pertinent."
"The cells are usually extracted from surgical biopsies and are subsequently isolated and cultured." Consider separating with a comma for better readability, e.g., "The cells are usually extracted from surgical biopsies, and are subsequently isolated and cultured."
"which will hopefully shape the future of endometriosis research." "which may shape the future of endometriosis research" is more concise and appropriate.
"With an estimated prevalence of 10%, it is thought to affect at least 190 million women of reproductive age worldwide." "It is estimated that 10% of women of reproductive age are affected, totaling at least 190 million worldwide" for clarity.
"However, recent genetic studies have theorized that epithelial progenitor cells likely play a key role in establishing ectopic nascent glands through the recruitment of polyclonal stromal cells and subsequent formation of infiltrating endometriosis lesions." This sentence is quite long and complex; breaking it into two sentences could improve clarity.
"This heterogenicity is of particular importance for slowly proliferating cells such as epithelial cells, as contamination with more quickly proliferating cells will have a large impact on the sample composition over time." "This heterogeneity is particularly important for slowly proliferating cells, such as epithelial cells, as contamination with rapidly proliferating cells can significantly impact sample composition over time."
Author Response
We have uploaded a detailed response as pdf document.

Reviewer 2 Report
Comments and Suggestions for Authors
In this review, the authors summarized in vitro and in vivo models of endometriosis and new approaches to modeling endometriosis.
I have some comments as follow:
1. In lines 46-49 it would be appropriate to make an introduction about what will be described below;
2. The authors can briefly described Table 1;
3. In line 143 it is preferable to write the entire name of MMP and TIMP, unless there is a glossary at the beginning of the article;
4. The authors can add some figures in the review instead of just mentioning the results.
Author Response

(The authors gave the same response as above.)

Reviewer 3 Report
Comments and Suggestions for Authors
Endometriosis, a chronic inflammatory condition affecting millions of women worldwide, is characterized by endometrium-like tissue outside the uterus. Despite its prevalence, the disease's pathophysiology remains poorly understood. This review highlights the significance of in vitro and in vivo models in advancing our understanding of endometriosis and explores emerging technologies, such as microfluidic cultures and 3D printing. These innovative approaches are poised to provide deeper insights into the disease's mechanisms and support the development of personalized diagnostic and therapeutic strategies.
1. Line 98 – Biomarker Specification for Sorting
The manuscript currently lacks clarity on the specific biomarkers used for cell sorting. It is critical for authors to specify these biomarkers to ensure reproducibility and to validate the relevance of their selected markers in isolating endometrial or endometriosis-specific cell populations. For example, are surface markers such as CD10 or EPCAM being used, or are molecular markers like HOXA10 or ERα involved? Adding this information would significantly enhance the credibility and scientific rigor of the methods section.
2. Mechanisms in Endometriosis Progression
The manuscript predominantly focuses on the expression changes (increases or decreases) of specific genes or proteins without delving into their mechanistic roles in endometriosis progression. I recommend that the authors provide a more detailed discussion of how these expression changes contribute to key processes such as inflammation, angiogenesis, extracellular matrix remodeling, or hormonal imbalances that characterize endometriosis. For instance, if VEGF or MMPs are implicated, their roles in neovascularization or tissue remodeling should be explicitly discussed, highlighting how these processes drive lesion establishment and persistence.
3. Content Depth and Focus
The description of in vitro and in vivo models of endometriosis in its current form is relatively broad and lacks depth, which detracts from the manuscript's impact. The authors should expand their discussion to emphasize how advanced models like 3D microfluidic cultures and organoids replicate the pathophysiological features of the disease, such as cellular interactions, hormone responsiveness, and the immune microenvironment. Similarly, for in vivo models, while rodents and primates are mentioned, the authors could discuss how novel strategies such as 3D printing or the use of menstrual blood-derived stromal cells are advancing the field. Moreover, the limitations of current models in fully replicating the human condition should be critically addressed, with suggestions for overcoming these gaps.
4. Extended Commentary on Diagnostic and Therapeutic Implications
The authors briefly mention how advances in modeling could improve diagnostics and therapeutics but fail to provide concrete examples or a forward-looking perspective. For instance, could these models be used for high-throughput drug screening, or might they reveal new biomarkers for early detection? Including such insights would enhance the clinical relevance of the manuscript.
Specific Recommendations
Line 98: Clearly list the biomarkers used for sorting and provide justification for their selection.
Mechanisms: Provide a comprehensive analysis of how key molecules and pathways contribute to the pathophysiology of endometriosis.
Models: Highlight how advanced models (e.g., organoids, 3D bioprinting) overcome limitations of traditional systems and discuss their translational potential.
Conclusion: Strengthen the conclusion by integrating insights on how these models can bridge the gap between basic research and clinical applications.
Author Response

(The authors gave the same response as above.)
